



**Boundary of nighttime ozone chemical equilibrium in the mesopause region: long-**
**term evolution from 20-year satellite observations**
Mikhail Yu. Kulikov[1], Mikhail V. Belikovich[1], Aleksey G. Chubarov[1], Svetlana O. Dementyeva[1], and
Alexander M. Feigin[1]
[1]A. V. Gaponov-Grekhov Institute of Applied Physics of the Russian Academy of Sciences, 46 Ulyanov
Str., 603950 Nizhny Novgorod, Russia
Correspondence to: Mikhail Yu. Kulikov (mikhail_kulikov@mail.ru)



**Abstract.** The assumption of nighttime ozone chemical equilibrium (NOCE) is widely used for retrieval of the Ox-HOx components in the mesopause from rocket and satellite measurements. In this work, recently developed analytical criterion of determining the NOCE boundary is applied (1) to study of connection of this boundary with O and H spatiotemporal variability basing on the 3D chemical transport modeling, and (2) to retrieve and analyze the spatiotemporal evolution of the NOCE boundary in 2002-2021 from SABER/TIMED data set. It was revealed, first, that the NOCE boundary well reproduces the transition zone dividing deep and weak diurnal oscillations of O and H at the low and middle latitudes. Second, the NOCE boundary is sensitive to sporadic abrupt changes in the middle atmosphere dynamics, in particular, due to powerful sudden stratospheric warmings leading to events of elevated (up to ~80 km) stratopause, which took place in January-February 2004, 2006, 2009, 2010, 2012, 2013, 2018 and 2019. Third, the space-time evolution of this characteristics expressed via pressure-height contains a clear signal of 11-year solar cycle in the range of 55ºS-55ºN. In particular, average annual the NOCE boundary averaged in this range of latitudes anticorrelates well with $F_{10.7}$ index with the coefficient of -0.96. Moreover, it shows a weak linear trend of 49.2±36.2 m/decade.

## 1 Introduction

The mesopause (80-100 km) is an interesting region of Earth' atmosphere possessing quite a number of unique phenomena and processes which can be considered as sensitive indicators/predictors of global climate change and anthropogenic influences on atmospheric composition (e.g., (Thomas et al., 1989)). Here, the temperature at middle and high latitudes in the summer reaches its lowest values (down to 100K (Schmidlin, 1992)). The temperatures below 150K lead to water vapour condensation and formation of the highest altitude clouds on Earth, the so-called Polar Mesospheric Clouds or Noctilucent Clouds, consisting primarily of water ice (Thomas, 1991). In opposite, the temperature of winter mesopause is essentially higher, so there is a strong negative temperature gradient between the summer and winter hemispheres. At these altitudes, atmospheric waves with various spatiotemporal scales are observed, in particular, internal gravity waves coming from the lower atmosphere. Destruction of gravity waves leads to strong turbulence that affects the atmospheric circulation and ultimately manifests itself in the mentioned temperature structure of this region.

Many layer phenomena in the mesopause are connected with the photochemistry of the $O_x$-$HO_x$ components (O, $O_3$, H, OH, and $HO_2$). Here, there is a narrow (in height) transition region where photochemistry behavior transforms rapidly from "deep" diurnal oscillations, when the difference between daytime and nighttime values of the $O_x$-$HO_x$ components can rich several orders of magnitude, to weak photochemical oscillations. As the result, above this region, there takes place O and H




accumulation and their layers formation manifesting in the appearance of a secondary ozone maximum
and airglow layers of OH and O excited states. Thus, $O_x$-$HO_x$ photochemistry in the mesopause is
responsible for the presence of important (first of all, from a practical point of view) indicators observed
in the visible and infrared ranges, which are widely used for ground-based and satellite monitoring of
climate changes and waves activity. Moreover, $O_x$-$HO_x$ photochemistry provides the total chemical
heating rate of this region, influences the radiative cooling and other useful airglows (for example, by $O_2$
excited states), involves in the plasma-chemical reactions and formation of the ionosphere layers. The
mentioned transformation of $O_x$-$HO_x$ behavior with height may happens via the nonlinear response of $O_x$-
$HO_x$ photochemistry to the diurnal variations of solar radiation in the form of subharmonic (with periods
of 2, 3, 4, and more days) or the chaotic oscillations (e.g., Sonnemann and Fichtelmann, 1997; Feigin et
al., 1998). This unique phenomenon was predicted many years ago (e.g., Sonnemann and Fichtelmann,
1987) and investigated theoretically by models with taking into account of different transport processes
(e.g., Sonnemann and Feigin, 1999; Sonnemann et al., 1999; Sonnemann and Grygalashvyly, 2005;
Kulikov and Feigin, 2005; Kulikov, 2007; Kulikov et al., 2020). It was revealed, in particular, the
appearance of nonlinear response is controlled by the vertical eddy diffusion (Sonnemann and Feigin,
1999; Sonnemann et al., 1999), so that 2-day oscillations can only survive at the real diffusion
coefficients, but the eddy diffusion in zonal direction leads to the appearance of so-called reaction-
diffusion waves in the form of propagating phase fronts of 2-day oscillations (Kulikov and Feigin, 2005;
Kulikov et al., 2020). Recently, the satellite data processing found the first evidence that 2-day
photochemical oscillations exist in the real mesopause (Kulikov et al., 2021).
While regular remote sensing measurements of most $O_x$-$HO_x$ components are still limited, the
indirect methods based on the physicochemical assumptions are useful tools to monitor these trace gases.
In many papers, the O and H distributions were retrieved from the daytime and nighttime rocket and
satellite measurements of the ozone and the volume emission rates of OH(v), O($^1$S), and $O_2(a^1\Delta_g)$ (e.g.,
Good, 1976; Pendleton et al., 1983; McDade et al., 1985; McDade and Llewellyn, 1988; Evans et al.,
1988; Thomas, 1990; Llewellyn et al., 1993; Llewellyn and McDade, 1996; Mlynczak et al., 2007, 2013a,
2013b, 2014, 2018; Smith et al., 2010; Xu et al., 2012; Siskind et al., 2008, 2015). The retrieval technique
is based on the assumption of the ozone photochemical/chemical equilibrium and physicochemical model
of corresponding airglow, which describe the relations between local O and H values and the
measurement data.
The daytime photochemical ozone equilibrium is good approximation everywhere in the
mesosphere - lower thermosphere (MLT) region (Kulikov et al., 2017) due to ozone photodissociation,
whereas the applicability of the assumption of nighttime ozone chemical equilibrium (NOCE) is limited:
there is an altitude boundary upper which NOCE is satisfied with accuracy better than 10%. Below this



boundary, the ozone equilibrium is disturbed essentially and cannot be used. Good (1976) supposed that
NOCE is fulfilled above 60 km, whereas other papers apply the NOCE starting from 80 km, independent
of latitude and season. However, studies of NOCE within the framework of the 3D chemical-transport
models (Belikovich et al., 2018; Kulikov et al., 2018a) revealed that the boundary of NOCE varies within
the range of 81–87 km, depending on latitude and season. Due to the practical necessity to determine the
local altitude position of this boundary, Kulikov et al. (2018a) presented a simple criterion localizing of
the equilibrium boundary using only the data provided by the SABER (Sounding of the Atmosphere
using Broadband Emission Radiometry) instrument onboard the TIMED (Thermosphere Ionosphere
Mesosphere Energetics and Dynamics). Using this criterion, Kulikov et al. (2019) retrieved annual
evolution of the NOCE boundary from the SABER data. It was revealed that the NOCE boundary
essentially depends on season and latitude and can rise up to ~ 86 km. Moreover, the analysis of the
NOCE boundary in 2003-2005 showed that this characteristic was sensitive to unusual dynamics of
stratospheric polar vortex during Arctic winter 2004, which was named as remarkable winter in the
50-year record of meteorological analyses (Manney et al., 2005). Moreover, Belikovich et al. (2018)
found by 3D simulation that the excited OH layer well repeats spatiotemporal variability of the NOCE
boundary. These results let us to speculate that the NOCE boundary can be considered as important
indicator of mesopause' processes.
The main goals of this paper are (1) to investigate the connection of the NOCE boundary according
the mentioned criterion with O and H variability with the use of the 3D chemical transport model, and (2)
to retrieve and analyze the spatiotemporal evolution of the NOCE boundary in 2002-2021 from
SABER/TIMED data set. In the next section, we present the used model. In Section 3, we describe
shortly the criterion to determine the NOCE boundary local height and study how this height relates with
features O and H distributions from the 3D model. Section 4 explains the methodology of determining the
NOCE boundary from satellite data. Section 5 presents the main results obtained from SABER/TIMED
data, which are discussed in Section 6.

## 2 3D model

We use the 3D chemical transport model of the middle atmosphere developed by the Leibniz
Institute of Atmospheric Physics (e.g., Sonnemann et al., 1998; Körner and Sonnemann, 2001;
Grygalashvyly et al., 2009; Hartogh et al., 2004, 2011). The three-dimensional fields of temperature and
winds were adopted by Kulikov et al. (2018b) from the Canadian Middle Atmosphere Model (Scinocca et
al., 2008) with an updated frequency of 6 hours. To exclude unrealistic jumps in the evolution of
calculated chemical characteristics, a linear smoothing between two subsequent updates of these





parameters is applied. The model takes into account 3D advective transport and vertical diffusive
transport (both, turbulent and molecular). The Walcek-scheme (Walcek, 2000) and the implicit Thomas
algorithm (Morton and Mayers, 1994) are used for advective and diffusive transport, respectively. The
model grid includes 118 pressure-height levels (from ground to ~135 km), 32 and 64 levels in latitude and
longitude, respectively. The chemical part considers 22 constituents (O, O($^1$D), $O_3$, H, OH, $HO_2$, $H_2O_2$,
$H_2O$, N, NO, $NO_2$, $NO_3$, $N_2O$, $CH_4$, $CH_2$, $CH_3$, $CH_3O_2$, $CH_3O$, $CH_2O$, CHO, CO, $CO_2$), 54 two- and
three-body reactions, and 15 photo-dissociation reactions. The model uses pre-calculated dependencies of
dissociation rates on the altitude and solar zenith angle (Kremp et al., 1999). For the chemistry
calculation, we apply the Shimazaki scheme (Shimazaki, 1985) at the integration time of 9 sec.

## 3 The NOCE criterion

The nighttime ozone chemistry at the mesopause heights is determined mainly by two reactions R1-
R2 (e.g., Allen et al., 1984), see Table 1. Thus, ozone equilibrium concentration ($O_3^{eq}$) corresponding to
the instantaneous balance between production and loss terms is as follows:
$$O_3^{eq} = \frac{k_1 \cdot O \cdot O_2 \cdot M}{k_2 \cdot H},$$     (1)
where $M$ is air concentration, $k_{1-2}$ are the corresponding rate constants of reactions (see Table 1).
As mentioned above, the NOCE criterion was developed in Kulikov et al. (2018a). The main idea is
that the local values of $O_3$ and $O_3^{eq}$ are close to each other ($O_3(t) \approx O_3^{eq}(t)$), when $\tau_{O_3} \ll \tau_{O_3^{eq}}$, where
$\tau_{O_3}$ is the ozone lifetime and $\tau_{O_3^{eq}}$ is the local time scale of $O_3^{eq}$:
$$\tau_{O_3} = \frac{1}{k_2 \cdot H},$$     (2)
$$\tau_{O_3^{eq}} \equiv \frac{O_3^{eq}}{|dO_3^{eq}/dt|} = \frac{O}{H \cdot \left| \frac{d}{dt}\left(\frac{O}{H}\right) \right|},$$     (3)
As shown in Kulikov et al. (2018a), $\tau_{O_3^{eq}}$ can be determined from a simplified photochemical model
describing the $O_x$-$HO_x$ evolution in the mesopause region (Feigin et al., 1998), so the criterion for validity
of the NOCE can be written in the form:
$$Cr = \frac{\tau_{O_3}}{\tau_{O_3^{eq}}} = 2\frac{k_1 \cdot k_4 \cdot O_2^2 \cdot M^2}{k_2}\left(1 - \frac{k_5 + k_6}{k_3}\right) \cdot \frac{1}{k_2 \cdot H \cdot O_3} \ll 1.$$     (4)
where $k_i$ are the corresponding reaction constants from Table 1. Calculations with the global 3D
chemistry-transport model of the middle atmosphere showed (Kulikov et al. 2018a) that the criterion
$\tau_{O_3}/\tau_{O_3^{eq}} \le 0.1$ well defines the boundary of the area where $|O_3/O_3^{eq} - 1| \le 0.1$.





Kulikov et al. (2023) presented the theory of chemical equilibrium of a certain trace gas $n$. Strictly
mathematically, the cascade of the sufficient conditions for $n_i(t) \cong n_i^{eq}(t)$ was derived considering its
lifetime, equilibrium concentration and time dependences of these characteristics. In case of the nighttime
ozone, it was proved that the $\tau_{O_3}/\tau_{O_3}^{eq} \ll 1$ is the main condition for NOCE validity and the criterion
$\tau_{O_3}/\tau_{O_3}^{eq} \leq 0.1$ limits the possible difference between $O_3$ and $O_3^{eq}$ to be no more than ~10%. Moreover,
Kulikov et al. (2023) slightly corrected the expression for the criterion (4):
$\text{Cr} = 2\frac{k_1 \cdot O_2 \cdot M}{k_2}\left(k_4 \cdot M \cdot O_2 \cdot \left(1 - \frac{k_5 + k_6}{k_3}\right) + k_2 \cdot O_3\right) \cdot \frac{1}{k_2 \cdot H \cdot O_3} \leq 0.1.$    (5)
Other important condition for $O_3 \approx O_3^{eq}$ at the time moment $t$ is:
$e^{\int_{t_{bn}}^{t} \tau_{O_3}^{-1} dt} \gg 1,$    (6)
where $t_{bn}$ is the time of the beginning of the night. It means the nighttime data measured near the sunset
should be excluded from consideration. Kulikov et al. (2023) revealed that, at the solar zenith angle $\chi >$
95°, the condition (6) is fulfilled in almost all cases.
Figures 1-3 demonstrate model examples of O and H time-height variations above different points
in three months. In order to focus our attention on diurnal oscillations, the concentrations are normalized
by mean daily values, correspondingly. One can see in all panels of these Figures, first, below 81-87 km,
"deep" diurnal oscillations occur. Due to the shutdown of sources at night and high rates of the main $HO_x$
and O sinks nonlinearly dependent on air concentration (Konovalov and Feigin, 2000), the variables
change during each night within a range of several orders of magnitude with low values of times
evolution. Above 83-88 km, the situation differs essentially from the previous case. One can see the
relatively weak diurnal oscillations. These regimes of O and H behavior are in consistent each other, i.e.
deep H diurnal oscillations correspond to the same dynamics in O and so on. There exists a few km thick
layer (transition zone) dividing deep and weak oscillations which height position is depended on latitude
and season. In particular, summer middle latitude transition is higher than in winter. Figures 1-3 show
also the magenta lines pointing the NOCE boundary in accordance to criterion (5) (Cr = 0.1). One can
see that the NOCE criterion almost perfectly reproduces the features of transition zone. Thus, our
criterion is not only the useful technical characteristic to retrieve O from satellite data, but it points the
important dynamical process in $O_x$-$HO_x$ photochemistry.

**4 Boundary of the NOCE from satellite data**
We use the version 2.0 of the SABER data product (Level2A) for the simultaneously measured
height profiles of temperature (T), $O_3$ (at 9.6 μm), and total volume emission rates of OH* transitions at



2.0 ($VER$) within the 0.0001–0.02 mbar pressure interval (altitudes approximately 75–105 km) in 2002-
2021. We consider only nighttime data when the solar zenith angle $\chi > 95°$.
Kulikov et al. (2018a) noted that the term $k_2 \cdot H \cdot O_3$ in the expression for the NOCE criterion can
be rewritten in the form depended on measurable characteristics only with the use of the corresponding
OH($v$) model by Mlynczak et al. (2013a):
$k_2 \cdot H \cdot O_3 = VER/A(T,M,O),$ (7)
where $A(T,M,O)$ is a function in square brackets of equation (3) in the paper by Mlynczak et al. (2013a)
with parameters corrected by Mlynczak et al. (2018):
$A(T,M,O) = 0.47 \cdot 118.35/(215.05 + 2.5 \cdot 10^{-11} \cdot O_2 + 3.36 \cdot 10^{-13} \cdot e^{220/T} \cdot N_2 + 3 \cdot 10^{-10} \cdot$
$O) + 0.34 \cdot 117.21/(178.06 + 4.8 \cdot 10^{-13} \cdot O_2 + 7 \cdot 10^{-13} \cdot N_2 + 1.5 \cdot 10^{-10} \cdot O) + 0.47 \cdot$
$117.21/(215.05 + 2.5 \cdot 10^{-11} \cdot O_2 + 3.36 \cdot 10^{-13} \cdot e^{220/T} \cdot N_2 + 3 \cdot 10^{-10} \cdot O) \cdot (20.05 + 4.2 \cdot$
$10^{-12} \cdot O_2 + 4 \cdot 10^{-13} \cdot N_2)/(178.06 + 4.8 \cdot 10^{-13} \cdot O_2 + 7 \cdot 10^{-13} \cdot N_2 + 1.5 \cdot 10^{-10} \cdot O).$
This function is the result of combination of the equations of physicochemical OH* balance in the $v = 8$
and $v = 9$ states. It depends on the constants of the processes describing sources and sinks on the
corresponding levels, in particular, the OH($v$) removal in collisions with $O_2$, $N_2$ and O. Below 86-87 km,
$A(T,M,O) \cong A(T,M,O = 0) \equiv A(T,M)$ because of relativity small O concentrations. Thus, combining
Eqs. (5) and (7), the NOCE criterion for SABER data can be recast in the following form:
$VER \geq VER_{min}(T,M) = 20 \cdot \frac{k_1 \cdot O_2 \cdot M}{k_2} (k_4 \cdot O_2 \cdot M \cdot \left(1 - \frac{k_5 + k_6}{k_3}\right) + k_2 \cdot O_3) \cdot A(T,M)$ (8)
Due to the strong air concentration-dependence, $VER_{min}$ decreases rapidly with height. In
particular, at 105 km, $VER \gg VER_{min}$. At 75 km, the relationship is the inverse. We determine the local
position of the NOCE boundary (pressure level $p_{eq}$ and altitude level $z_{eq}$) according to the criterion (8),
where  $VER = VER_{min}(T,M)$. We carried out special verification that approximation $A(T,M,O) \cong$
$A(T,M)$ is valid near the NOCE boundary. With the use of annual SABER data, we calculated
simultaneous datasets of $A(T,M)$ and $A(T,M,O)$. In second case, we used retrieved O from the same
SABER data. The maximum and mean differences between $A(T,M)$ and $A(T,M,O)$ were found to be ~
2% and ~ 0.1%, respectively.
The total range of latitudes according to the satellite trajectory over a month is ~(83.5$^°$S - 83.5$^°$N).
This range was divided into 20 bins and all single values of $p_{eq}$ and $z_{eq}$ falling into one bin during a
month were averaged, respectively. For convenience, mean values of $p_{eq}$ were recalculated into pressure-
heights (pseudoheights) $z_{eq}{}^{pa}$. The dependence of $z_{eq}{}^{pa}$ on the pressure was taken from Mlynczak et al.
(2013a, 2014).





Kulikov et al. (2023) studied the systematic uncertainty of the retrieved NOCE boundary height.
Following the typical analysis presented, for example, in Mlynczak et al. (2013a, 2014), the uncertainty
was obtained by calculating the root-sum-square of the individual sensitivity of the retrieved
characteristics to the perturbation of $O_3$, T, rates of reactions, and parameters of the $A$ function. The
systematic error of $z_{eq}^{pa}$ and $z_{eq}$ varies in the range of 0.1-0.3 km, whereas the random error is negligible
due to averaging in time and space.

## 5 NOCE boundary in 2002-2021 from SABER/TIMED data: main results

Figures 4-7 demonstrate the contour map of space-time evolution of pseudoheight $z_{eq}^{pa}$ in 2002-
2021 and examples of $z_{eq}^{pa}$ time evolution, mean annual cycle and Fourier spectra at different latitudes.
It can be seen, first, above ~55ºS,N, there are data gaps due to the satellite sensing geometry. For
example, at 66.8-75.15°S,N in 2002-2014, measurements cover 6 months per year only. In 2015, because
of slight change in satellite geometry, additional months appeared. This is especially noticeable above
~66ºS,N and manifests itself by extension of the variation range of $z_{eq}^{pa}$ at these latitudes in 2015-2021.
Second, the variation range of $z_{eq}^{pa}$, annual cycle and spectrum of harmonic oscillations depend
essentially on latitude. Near the equator, $z_{eq}^{pa}$ varies in the 81-83 km range mainly and there are two
main harmonics with periods of 1/2 and 1 year in the spectrum. At low latitudes, the variation range of
$z_{eq}^{pa}$ narrows down to a minimum (~82.2-83.2 km at 16.7-20.05°S,N) that is accompanied by
appearance of wide spectrum of harmonics with periods of 1/5, 1/4, 1/3, 1/2, and 1 year. At middle
latitudes, the range of $z_{eq}^{pa}$ variation monotonically increases up to ~81-85.5 km with latitude and the
harmonic with period of 1 year becomes the main mode in the spectrum of oscillations. At both, low and
middle latitudes, there is no signal from quasi-biennial oscillations but one can see remarkable amplitude
of harmonic with a period of ~10 years, which can be associated with manifestation of 11-year solar
cycle. Note mentioned features are typical for both hemispheres. At high latitudes, $z_{eq}^{pa}$ varies in the
range of 79-86 km. At these latitudes, it can seen the main difference between north and south
hemispheres: the sharp falls and rises of the north boundary of NOCE by several km (up to 3-4 km)
appearing in January-February 2004, 2006, 2009, 2010, 2012, 2013, 2018 and 2019 and absenting at
south latitudes.
Analyzing the Figure 6, one can note the following redistribution in the annual cycle with
increasing latitude from equator to polar latitudes. Near the equator, the annual cycle has two maxima in
June – July and in December – January. The first one is more pronounced. That is why there are two main
harmonics with periods of 1/2 and 1 year in the spectrum. At low latitudes, one maximum (summer)





remains in place, and the other begins to approach the first. As the result, the wide spectrum of harmonics
takes place. At middle latitudes, the maxima gradually merge so that the 1 year-harmonic becomes the
main.
Figures 8-9 demonstrate the contour map of space-time evolution of average annual $z_{eq}^{pa}$
($< z_{eq}^{pa} >$, hereafter, the angle brackets are used to denote the values averaged in time and space) in
2002-2021 and examples of time evolution of this characteristic at different latitudes. Basing on Fourier'
spectra presented in Figure 7, we can suppose that, at low and middle latitudes, the interannual variation
of $< z_{eq}^{pa} >$ is caused by 11-year solar cycle mainly. Figure 10 presents the correlation coefficient of
$< z_{eq}^{pa} >$ with $F_{10.7}$ index (solar radio flux at 10.7 cm, see red curve in Figure 8) as a function of
latitude. One can see good anticorrelation (with coefficient from -0.74 to -0.9) between ~55ºS and ~55ºN.
At high latitudes, the absolute value of correlation coefficient decreases sharply up to 0.56 in the south
and 0.1 in the north. Blue curve in Figure 11 shows latitude-averaged $< z_{eq}^{pa} >$ in the range of 55ºS-
55ºN. In this case, the anticorrelation with $F_{10.7}$ index is close to ideal (coefficient ~ -0.96).
With the use of multiple linear regression:
$$< z_{eq}^{pa} > (year) = const + \alpha \cdot year + \beta \cdot F_{10.7}(year), \quad (9)$$
we determined slow linear trend in $< z_{eq}^{pa} >$ as a function of latitude in the range of 55ºS-55ºN (see
Figure 12). One can see a tendency to increase $< z_{eq}^{pa} >$ at most latitudes with trend up to 10 m/year,
but with high uncertainty. Applying the regression analysis to latitude-averaged $< z_{eq}^{pa} >$ (blue curve in
Figure 11) gives us a more statistically significant value of the trend: 4.92±3.62 m/year.
Figures 13-16 demonstrate the contour map of space-time evolution of real altitude of NOCE
boundary in 2002-2021, examples of $z_{eq}$ time evolution, mean annual cycle and the Fourier spectra at
different latitudes. Comparing with Figures 4-7, it can be seen, first, $z_{eq}$ repeats many qualitative features
of space-time evolution of $z_{eq}^{pa}$. In particular, in the direction from the equator to the poles, the variation
range of $z_{eq}$ first decreases up to 1 km at 16º-25ºS,N, then expanding to several km at middle and high
latitudes. In Figure 15, one can see the same redistribution of the annual cycle with latitude, as it was
mentioned in Figure 6. Near the equator, the annual cycle possesses two maxima occurring in June – July
and in December – January. At low latitudes, one maximum continues to be in summer, whereas the other
shifts into the spring. At middle latitudes, the maxima gradually coalesce forming a single summer
maximum. At north high latitudes, there are the same local sharp variations of the NOCE boundary in
January-February 2004, 2006, 2009, 2010, 2012, 2013, 2018 and 2019, which absent at south latitudes.
Second, on average, $z_{eq}$ is lower than $z_{eq}^{pa}$. The difference $z_{eq}^{pa} - z_{eq}$ varies in the range of 0 - 1.5 km



at equator, $0 - 2.5$ km at $50°$-$58°$S,N and $1$-$4$ km at $75°$-$83°$S,N. The maxima and minima of ${z_{eq}}^{pa} - z_{eq}$
are reached in winter and summer, respectively. In general, the variation range of $z_{eq}$ during the year is
wider by about 1.5-4 km depending on latitude. Third, the spectra of harmonic oscillations are similar to
${z_{eq}}^{pa}$ spectra except for no signal of the 11-year solar cycle.

266        Figures 17-18 demonstrate the contour map of space-time evolution of average annual $z_{eq}$

($< z_{eq} >$) in 2002-2021 and examples of time evolution of this characteristics at different latitudes. One
can see, at all latitudes, there is no clear evidence of 11-year solar cycle manifestation. This is confirmed
by the calculation of the correlation coefficient of $< z_{eq} >$ with $F_{10.7}$ index as a function of latitude (see
Figure 19). Moreover, the latitude-averaged (in the range of $55°$S-$55°$N) $< z_{eq} >$ has a correlation
coefficient equal to -0.54.

272        With the use of multiple linear regression as in the case of $< {z_{eq}}^{pa} >$, we determined slow linear

trend in $< z_{eq} >$ as a function of latitude in the range of $55°$S-$55°$N (see Figure 20). One can see a
tendency to decrease $< z_{eq} >$ at most latitudes with trend up to -10 m/year, but with high uncertainty.
Applying the regression analysis to latitude-averaged $< z_{eq} >$ gives us the trend equal to -4.48±6.73
m/year.

## 6 Discussion

279        The NOCE boundary is important technical characteristics for the correct application of NOCE

approximation to retrieve the nighttime distributions of minor chemical species of MLT. Remind also,
that Belikovich et al. (2018) found by 3D simulation that the excited hydroxyl layer well repeats
spatiotemporal variability of the NOCE boundary. Let discus the obtained results from the point of view
of other possible applications of this feature.

284        The carried out analysis revealed unusual behavior of the NOCE boundary at the north pole

latitudes in January-February 2004, 2006, 2009, 2010, 2012, 2013, 2018 and 2019. All these time periods
are known for strong local changes of the middle atmosphere dynamics due to extremely powerful sudden
stratospheric warming which leaded to appearance of elevated (from typical 50-60 km to ~80 km)
stratopause events (see García-Comas et al. (2020) and references there). Thus, we can speculate that the
boundary of NOCE is sensitive to sporadic abrupt changes in the dynamics of the middle atmosphere.

290        The space-time evolution of the NOCE boundary expressed in terms of pseudoheights contains a

clear signal of the 11-year solar cycle in the range of $55°$S-$55°$N, which is suppressed mainly at high



latitudes. The weak correlation of $z_{eq}{}^{pa}$ with $F_{10.7}$ index at south high latitudes may be caused by the
mentioned data gaps due to satellite sensing geometry. The same reason and distortions by sudden
stratospheric warming, evidently, determine no correlation at north high latitudes. Thus, $z_{eq}{}^{pa}$ at low and
middle latitudes can be considered as sensitive indicator of solar activity. The analysis of reasons why
solar cycle does not manifest itself in spatio-temporal variability of $z_{eq}$ requires a separate study and is
beyond the scope of this work.

298         Figures 6 and 15 present interesting peculiarity. At middle latitudes, summer $z_{eq}{}^{pa}$ and $z_{eq}$ are

remarkably (for several kilometers) higher than winter ones, while the opposite relationship could be
expected. Due to more effective daytime $HO_x$ photoproduction at these altitudes, summer H values at the
beginning of nights are higher than the ones in winter. So, the summer ozone lifetimes should be less and
condition of NOCE is more favorable than in winter. Nevertheless, the same ratio between summer and
winter the NOCE boundaries at middle latitudes was revealed in Belikovich et al. (2018) and Kulikov et
al. (2018a), where the boundary of this equilibrium was determined by direct comparison of $O_3$ and $O_3{}^{eq}$
concentrations from results of 3-D chemical-transport models. Based on results in Section 3, one can
assume that the discussed effect is connected with the height position of the transition zone, which
demonstrates the same variation (see Figures 1-3). Kulikov et al. (2023) derived the equations describing
pure chemical O and H nighttime evolution:
$$\begin{cases} \frac{dO}{dt} = -2 \cdot k_4 \cdot M \cdot O_2 \cdot H - 2 \cdot k_2 \cdot H \cdot O_3 \\ \frac{dH}{dt} = -2 \cdot k_4 \cdot M \cdot O_2 \cdot \frac{k_5+k_6}{k_3} \cdot \frac{H^2}{O} \end{cases}. \tag{10}$$

Neglecting the second term in the first equation as secondary, this system can be analytically solved, so
that nighttime evolution times of O and H are as follows:
$\tau_O \equiv \frac{O}{|dO/dt|} = \frac{1}{2 \cdot k_4 \cdot M \cdot O_2} \cdot \left(\frac{O}{H}\right)_{t=t_{bn}} - \left(1 - \frac{k_5+k_6}{k_3}\right) \cdot (t - t_{bn}),$  (11)
$\tau_H \equiv \frac{H}{|dH/dt|} = \frac{1}{2 \cdot k_4 \cdot M \cdot O_2} \cdot \frac{k_3}{k_5+k_6} \cdot \left(\frac{O}{H}\right)_{t=t_{bn}} - \left(\frac{k_3}{k_5+k_6} - 1\right) \cdot (t - t_{bn}),$  (12)
where $t_{bn}$ is the time of the beginning of the night, $\left(\frac{O}{H}\right)_{t=t_{bn}}$ is the ratio $O/H$ at the beginning of the
night. Note that $k_3$ is essentially larger than $k_5 + k_6$ (see Table 1). Basing on daytime O and H
distributions in mesopause region obtained in Kulikov et al. (2022), we calculated $O/H$ in summer and
winter. During the summer, this ratio at middle latitudes is remarkably less than in winter, whereas air
concentration increases due to a decrease in temperature. As the result, summer $\tau_O$ and $\tau_H$ are essentially
less their winter values that explain the summer rise of transition zone and the NOCE boundary.



**7 Conclusions**

322   The NOCE criterion is not only the useful technical characteristics to retrieve O from satellite data,
but it reproduces the transition zone position, which divides deep and weak diurnal oscillations of O and
H at low and middle latitudes.

325   The boundary of NOCE according the criterion is sensitive to sporadic abrupt changes in the
dynamics of the middle atmosphere.

327   The NOCE boundary at low and middle latitudes expressed in pseudoheight contains a clear signal
of 11-year solar cycle and can be considered as sensitive indicator of solar activity.

329   At middle latitudes, summer boundary of NOCE is remarkably (for several kilometers) higher than
winter one that is accompanied by the same variation of the transition zone. This effect is explained by
the markedly lower values of the O and H nighttime evolution times in summer than in winter due to
lower values of the ratio $O/H$ at the beginning of the night and air concentration increase.

**Data availability.** The SABER data are obtained from the website (https://saber.gats-inc.com). The data
of solar radio flux at 10.7 cm in 2002-2021 were downloaded from
http://www.wdcb.ru/stp/solar/solar_flux.ru.html and https://www.spaceweather.gc.ca/forecast-
prevision/solar-solaire/solarflux/sx-5-en.php.

**Code availability**. Code is available upon request.

**Author contributions.** MK and MB carried out the data processing and analysis and wrote the
manuscript. AC, SD, and AM contributed to reviewing the article.

**Competing interests.** The authors declare that they have no conflict of interest.

**Acknowledgements.** The authors are grateful to the SABER team for data availability.

**Financial support.** This work was supported by the Russian Science Foundation under grant No. 22-12-
00064 (https://rscf.ru/project/22-12-00064/) and state assignment no. 0729-2020-0037.



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





**Table 1.** List of reactions with corresponding reaction rates (for three-body reactions $[\text{cm}^6 \text{ molecule}^{-2}$
$\text{s}^{-1}]$, for two-body reactions $[\text{cm}^3 \text{ molecule}^{-1} \text{ s}^{-1}]$) taken from Burkholder et al. (2020).

|  | Reaction | Rate constant |
|---|---|---|
| **R1** | $O+O_2+M \rightarrow O_3+M$ | $k_1 = 6.1 \cdot 10^{-34}(298/T)^{2.4}$ |
| **R2** | $H+O_3 \rightarrow O_2+OH$ | $k_2 = 1.4 \cdot 10^{-10} exp(-470/T)$ |
| **R3** | $O+HO_2 \rightarrow O_2+OH$ | $k_3 = 3 \cdot 10^{-11} exp(200/T)$ |
| **R4** | $H+O_2+ M \rightarrow HO_2+M$ | $k_4 = 5.3 \cdot 10^{-32}(298/T)^{1.8}$ |
| **R5** | $H+HO_2 \rightarrow O_2+H_2$ | $k_5 = 6.9 \cdot 10^{-12}$ |
| **R6** | $H+HO_2 \rightarrow O+H_2O$ | $k_6 = 1.6 \cdot 10^{-12}$ |






Figure 1. O and H time-height variations above different points in January calculated by the 3D chemical transport model of the middle atmosphere. The concentrations are normalized by mean daily values, correspondingly The dark bars mark daytime, light bars mark nighttime. The magenta lines point the NOCE boundary in accordance to criterion (5) ($Cr = 0.1$).

516





Figure 2. The same as in Fig. 1, but in April.



Figure 3. The same as in Fig. 1, but in July.





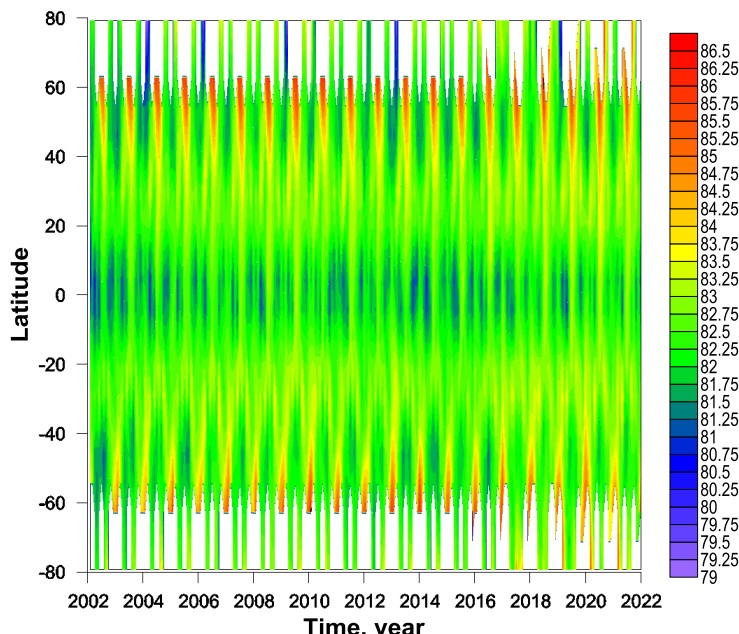


Figure 4. The space-time evolution of $z_{eq}{}^{pa}$. White color indicates data gaps due to the satellite sensing

geometry.











Figure 5. The time evolution of $z_{eq}{}^{pa}$ at different latitudes.



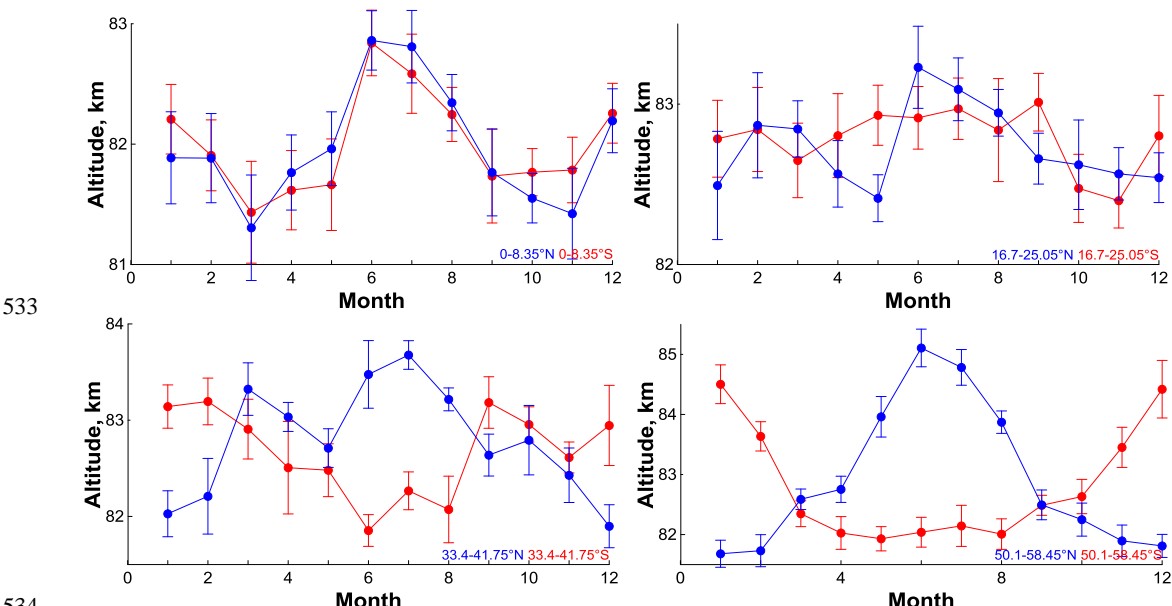



Figure 6. Average (for 2002-2021) annual cycle of $z_{eq}{}^{pa}$ at some latitudes.






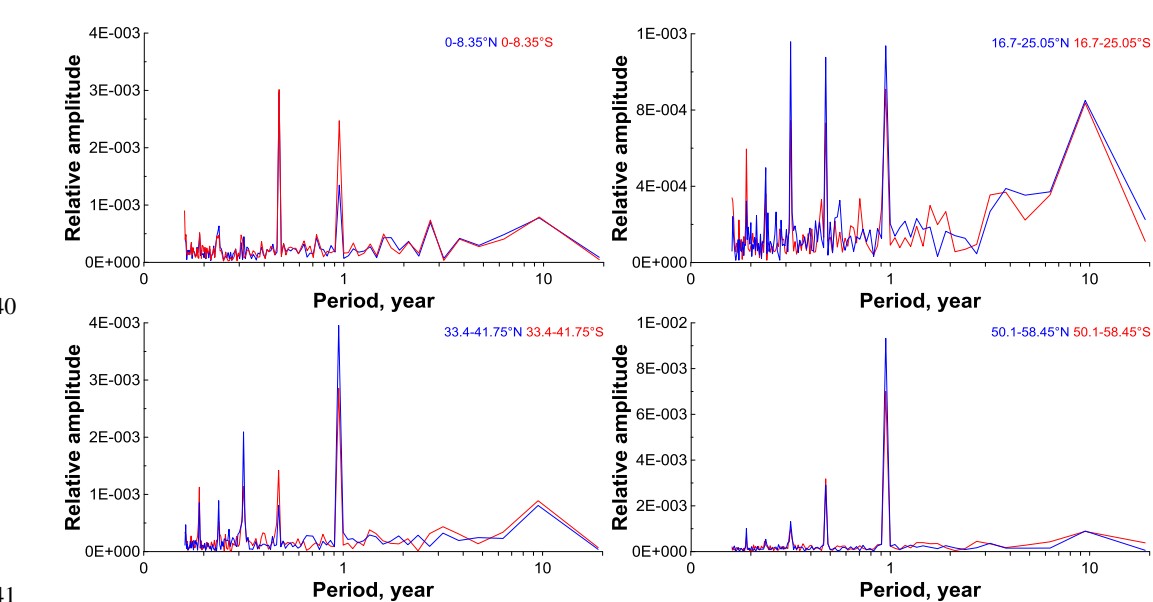



Figure 7. Fourier' spectra of $z_{eq}^{\ pa}$ time evolution at the same latitudes as in Figure 6. In each spectrum, the amplitudes of harmonics were normalized to corresponding zero harmonic.






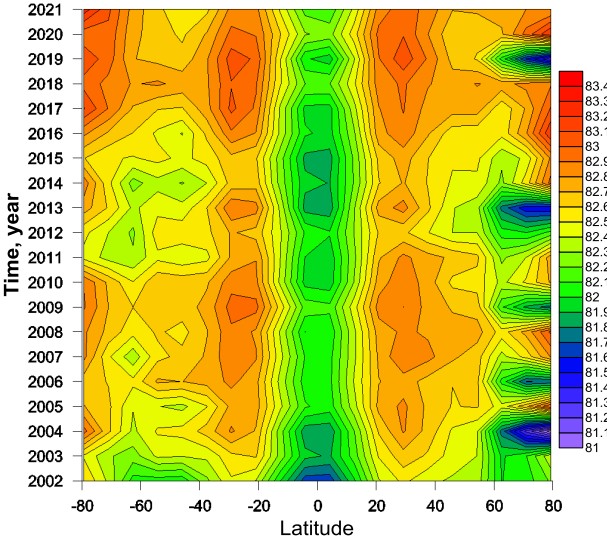


Figure 8. The latitude-time evolution of average annual $z_{eq}{}^{pa}$.









Figure 9. The time evolution of average annual $z_{eq}^{pa}$ ($< z_{eq}^{pa} >$) at different latitudes.





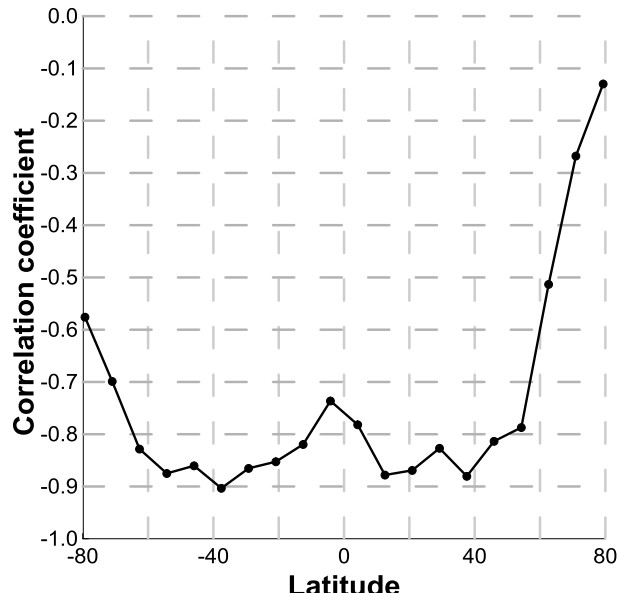


Figure 10. The correlation' coefficient of $< {z_{eq}}^{pa} >$ with $F_{10.7}$ index at different latitudes.




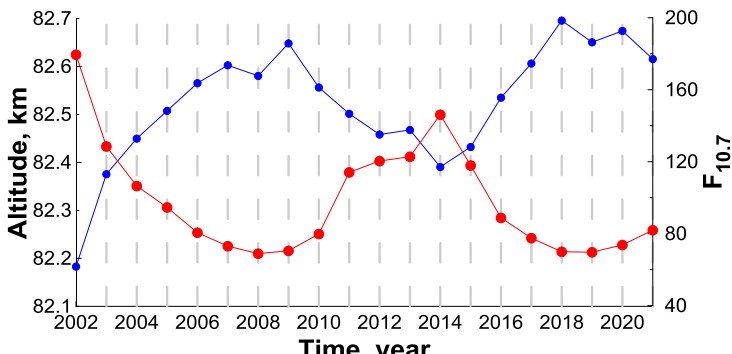



Figure 11. Red curve: $F_{10.7}$ index (solar radio flux at 10.7 cm). Blue curve: latitude-averaged $< z_{eq}{}^{pa} >$
in this range between ~55ºS and ~55ºN.






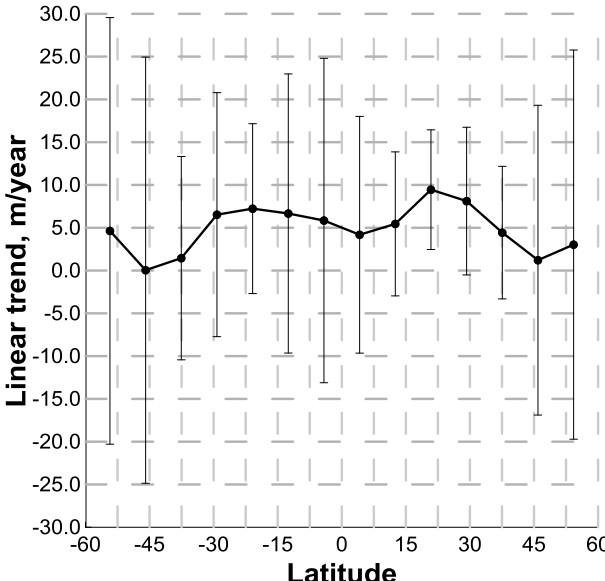


Figure 12. Linear trend in $< z_{eq}{}^{pa} >$ at different latitudes derived from multiregression analysis.





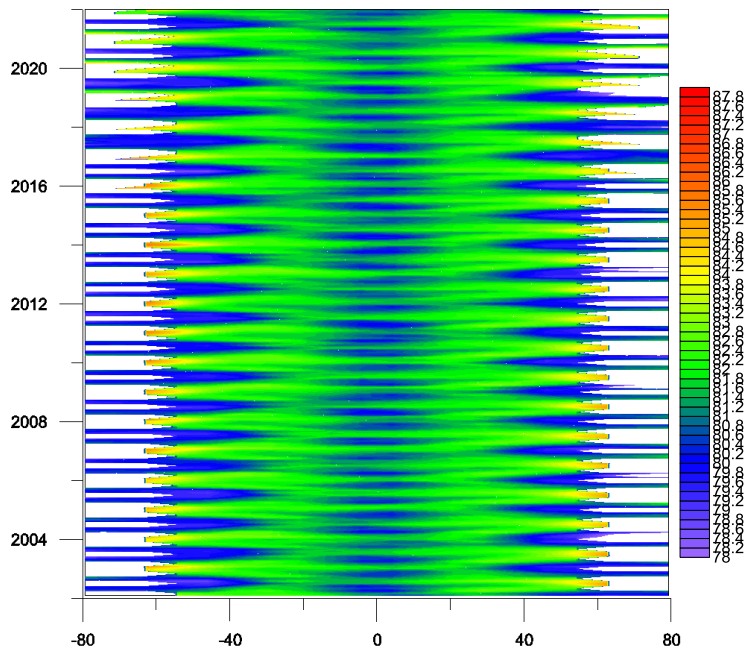


Figure 13. The space-time evolution of $z_{eq}$. White color indicates data gaps due to the satellite sensing

geometry.







Figure 14. The time evolution of $z_{eq}$ at different latitudes.



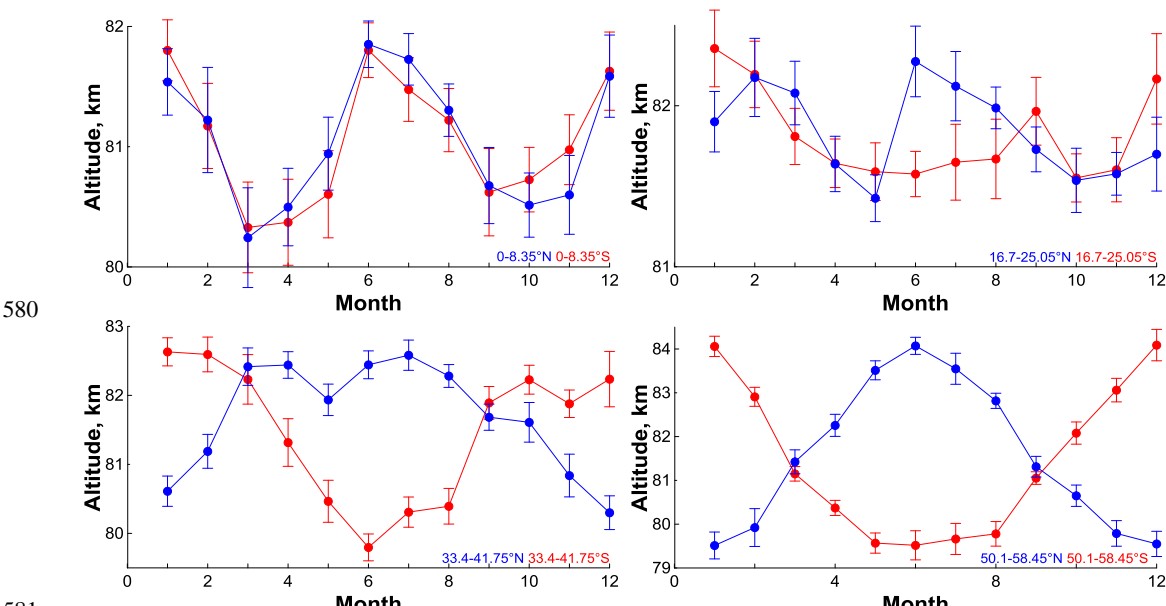

Figure 15. Average (for 2002-2021) annual cycle of $z_{eq}$ at some latitudes.





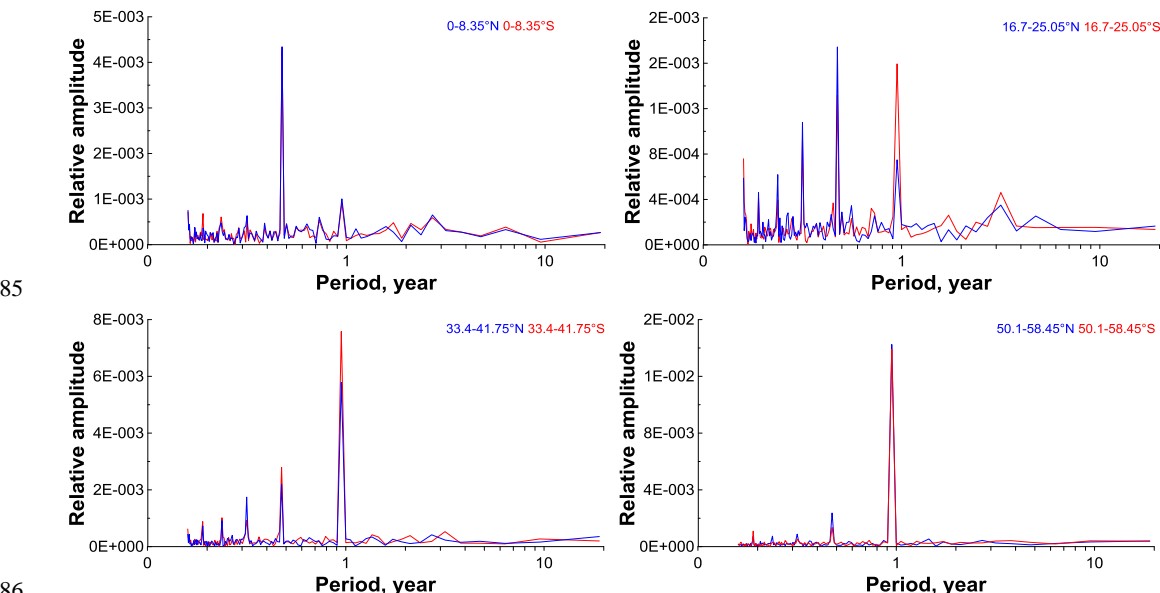

Figure 16. Fourier' spectra of $z_{eq}$ time evolution at different latitudes. In each spectrum, the amplitudes of

harmonics were normalized to corresponding zero harmonic.



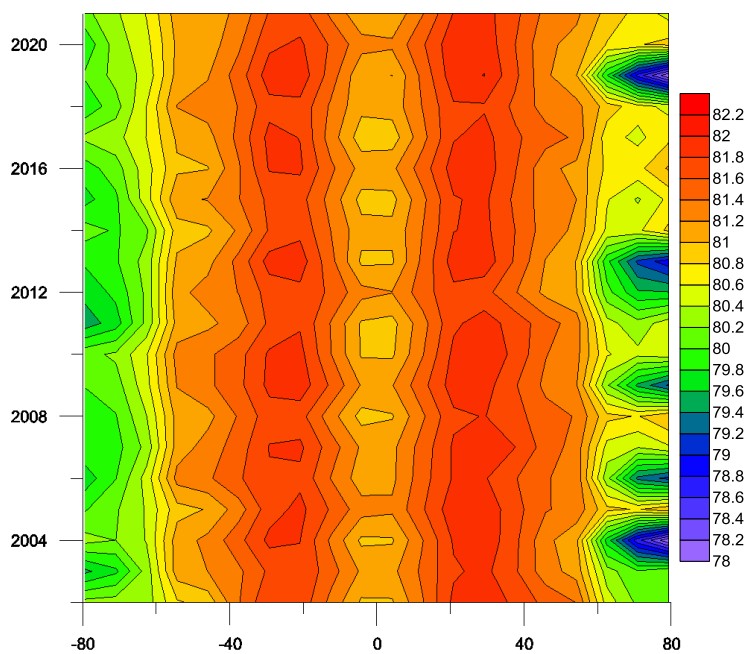

Figure 17. The latitude-time evolution of average annual $z_{eq}$.










Figure 18. The time evolution of average annual $z_{eq}$ ($< z_{eq} >$) at different latitudes.



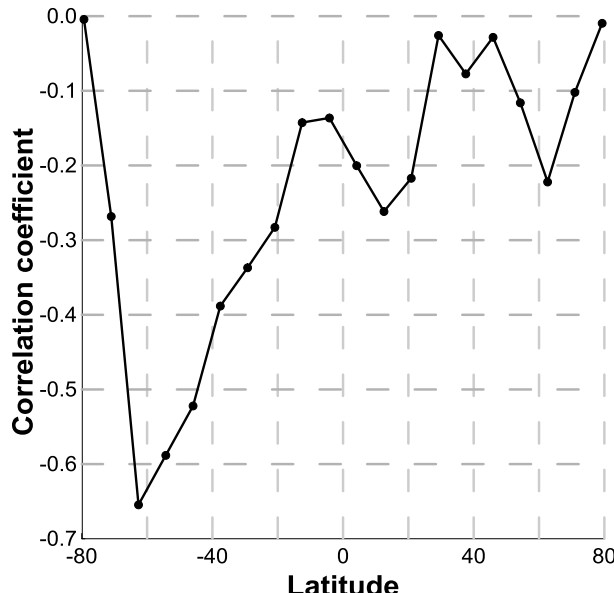


Figure 19. The correlation' coefficient of $< z_{eq} >$ with $F_{10.7}$ index at different latitudes.




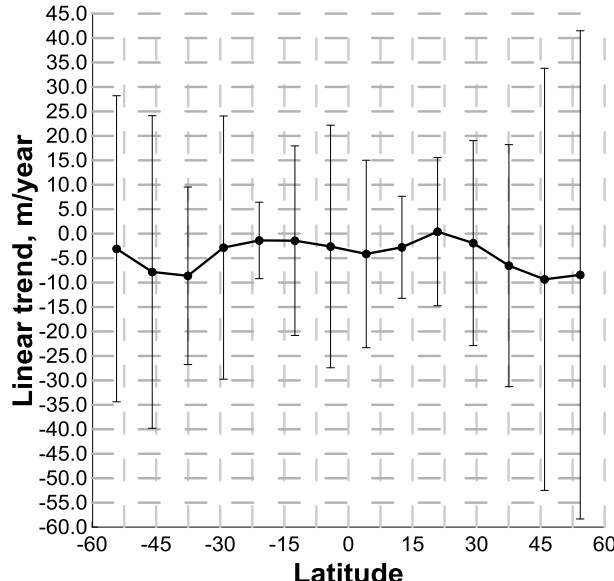


Figure 20. Linear trend in $< z_{eq} >$ at different latitudes derived from multiregression analysis.