# Peer review of "Boundary of nighttime ozone chemical equilibrium in the mesopause region: long- term evolution from 20-year satellite observations"

_EGUsphere, 2023_

## Referee Comment (RC2)

Review of "Boundary of nighttime ozone chemical equilibrium in the mesopause region: long2 term evolution from 20-year satellite observations" by Kulikov et al.

**Overview**

This paper presents a derivation of the nighttime ozone chemical equilibrium (NOCE) condition. The motivation for the paper is that minor atomic species O and H are key elements of the energy budget and chemistry of the mesopause region. These species have no readily/easily observable features to enable their direct observation by satellites or from the ground. Consequently, the approach used to derive these species is to infer them based on chemical relations between species such as ozone and observations of radiative emissions from the hydroxyl radical. The assumption behind these approaches is that ozone and the species H and O are in chemical equilibrium with each other at night.

The paper derives conditions under which the equilibrium assumptions would be true and assigns the upper altitude limit of these as the 'NOCE boundary.' The NOCE boundary is derived in depth based on observations made by the SABER instrument for the 20-plus year length of the mission. The NOCE boundary is analyzed in terms of altitude, latitude, and time. The NOCE boundaries are also examined for correlation with the 11-year solar cycle and for trends.

The paper presents interesting considerations for the derivation of O and H in the mesopause region from satellite observations. These considerations are important for the ongoing SABER instrument on the TIMED satellite and for analysis of data from the SCHIAMACHY instrument that formerly operated on EnviSAT.

**Recommendation**

The paper is returned for major revision. Below in the "Comments" section I list a number of comments for the authors to address. These are in no order of importance but rather they are chronological according to the paper.

There is one major weakness that I have for the entire paper – despite all the analyses of the NOCE boundary, and the apparent demonstration that ozone is not in chemical equilibrium over much of the mesopause region, the paper makes no statement as to how current the assumption of chemical equilibrium by SABER and SCHIAMCAHY affects the quality and the uncertainty in those datasets. Do they need reprocessing? How would that be done? What should current users assume for their analyses using these data? A revised paper must answer these questions.

Another major weakness in the paper is the trend analysis presented in Figures 12 and 20. These and their accompanying analyses and discussions must be deleted. The uncertainties (1-sigma) are larger than the derived trend, indicating that the trend is insignificant.

Also, the distinction between $z_{eq}$ and $z_{eq}^{pa}$ is not clear nor is the approach for deriving $z_{eq}^{pa}$. The paper needs to examine and clarify if these really are different parameters and why they both need to be analyzed to understand NOCE from a perspective of retrieving O and H from satellite observations.

The authors are also given several items in the Comments regarding the analysis and interpretation of the SABER data to address.

**Comments**

Line 40 – I believe the authors intend the word 'reach' instead of 'rich'.

Line 93-94 – It seems a word is missing between "according" and "the mentioned". Perhaps the authors mean "according to the mentioned"?

Line 120-124. The authors neglect loss of ozone by reaction with atomic oxygen in the mesopause region. At the level of which they appear to be investigating the chemistry, this process should be included. Table 1 confirms that this reaction is not considered.

Line 146-148. The authors need to specify which 'nighttime data' need to be 'excluded from consideration". O3, H, O, all?

Figures 1-3. The magenta line is very difficult to see in the figures unless they are substantially enlarged on the screen. It is also confusing with the large areas of similar colors above about 85 km in all figures. Perhaps a black line would be more visible? The caption should also state the meaning of the Cr = 0.1 condition/boundary to facilitate the interpretation of the figure.

Lines 170-185. The analysis here must be re-worked and compared with the model and derivation of O and H reported by Panka et al., including the results of H and OH reported in the literature. To date there has been no extensive comparison of the Panka et al. [O] results with the Mlynczak et al., 2018 results. The Panka et al. approach appears to more rigorously include the recent discovery of the importance of collisions with $O(^1D)$. The Panka data are hosted on the SABER website.

https://agupubs.onlinelibrary.wiley.com/doi/10.1029/2018GL077677

https://agupubs.onlinelibrary.wiley.com/doi/10.1029/2020GL091053

Line 191 – First full sentence, "In *the* second case.."

Line 194-198, and the subsequent description of Figures 4, 5, 6, and 7. The exact altitude of each pressure level is included in the SABER database. There is no need to approximate these altitudes based on a small subset of approximate altitudes and pressures mentioned in the Mlynczak 2013a and 2014 references. These altitudes are derived by SABER consistent with hydrostatic balance and the very accurate pointing of the SABER instrument. As presented in the paper, there could be substantial error in the assignment of altitude based on the method of altitude assignment as described by the authors.

Line 198. The authors should also specify, again to remind the reader, the meaning of $p_{eq}$ and $z_{eq}^{pa}$, and state this in the Figure captions.

Figure 6. The comparisons shown are for calendar months. The paper mentions 'changes in satellite geometry'. These 'changes' are such that the local time sampled by the spacecraft (and

hence, by SABER) is does not remain constant in a given calendar month over the course of the mission. It is slowly drifting. Thus, the local times sampled in every "January" are not the same over the course of the mission. Please see:

https://agupubs.onlinelibrary.wiley.com/doi/10.1029/2018JA025892

This fact suggests that the results in Figure 6 and any other Figure or results involving monthly calculations or comparisons may be incorrect.

Figure 7. Please mark the periods less than one year on the x-axis to make it easier for the reader to discern these. What is the statistical significance of these features? Have the authors considered a Lomb periodogram which provides a significance test for the derived periodic features?

Lines 234-243. The paper describes anticorrelation between average $z_{eq}^{pa}$ and the 11-year solar cycle. The authors should provide some rationale for why the altitude/pressure at which night ozone is (or is not) in photochemical equilibrium depends on the slowly varying solar cycle. Specifically, what atmospheric characteristics cause this? Is it a feedback from temperature and perhaps the 'breathing' or expansion and contraction of the atmosphere with the cyclic heating and cooling of the solar cycle?

Line 244-249. It is assumed that the uncertainties plotted in Figure 12 are 1-sigma values. All of the 1-sigma uncertainties are larger than the derived trend value, except at 20 N. At 2-sigma, all of them are. These results and figure 11 should be removed from the paper as the significance of them is marginal at best.

Figure 13. The color bar is very difficult to read as the altitudes all run together. It is also very difficult to discern anything quantitative about the annual variations shown in the figure.

Figure 14. Although there are differences in Figures 5 and 14 ($z_{eq}^{pa}$ and $z^{eq}$, respectively), it is not clear what these differences are trying to show, or if there is a real difference. See my previous comment about the assignment of altitude to $z_{eq}^{pa}$. For clarity, SABER's natural vertical coordinate is pressure. All the data are retrieved as a function of pressure. There is no 'assignment' of pressure. Similarly, the temperature retrieval profile that is in hydrostatic equilibrium, the altitudes assigned to the pressure surfaces are accurate, and are derived in part from the accurate knowledge of the position of the field of view of the instrument as is scans the limb.

Figures 17-18. Same comment as for Figure 14.
Figure 19 and 20. These results and Figures should be deleted. The trends in Figure 20 are not significant even at the 1-sigma level.

Line 290-297. The reason for the discrepancy noted in the $z_{eq}$ and $z_{eq}^{pa}$ may have a simple explanation, as noted above. It is not clear how the authors assigned $z_{eq}^{pa}$, as noted above.

---

## Author Comment (AC1)

Dear Editor,

We would like to say many thanks the Referee for taking the time to review our manuscript and valuable recommendations. We have tried to follow the referee remarks and have utilized all of them.

In the following, we address the comments point by point and show how the manuscript has been changed accordingly to the comments. Below comments by Referees are in red, our responses are in black, the changes in the manuscript are in blue and brackets.

**Response to the comments on the paper by Referee 1**

The title does not accurately reflect the essence of what has been done in this article. One might get the impression that the equilibrium boundary was taken from satellite data. In reality, this characteristic was found using SABER data. Should be corrected, for example: "...: long-term evolution of the boundary determined (OR derived) using 20-year satellite observations".

The title has been corrected to "**Boundary of nighttime ozone chemical equilibrium in the mesopause region: long-term evolution determined using 20-year satellite observations** ".

I would like to note that the nighttime ozone equilibrium boundary was investigated in previous papers of the authors using 3D MLT modelling. There the criterion was also proposed, which in this work is applied to determine the altitude position of this boundary, using already the data of real measurements. It is well known that any model is an idealized representation of the real reality and, in principle, may not take into account some important features of the natural object, poorly formulated mathematically. In this regard, a very important question that needs to be clarified at least in the discussion. Can the authors present any other indications of the nighttime ozone equilibrium boundary in the SABER profiles or in the O and H profiles reconstructed from these data?

In the Discussion of revised manuscript, we added short discussion concerning this recommendation (see lines 386-392):

"Finally, let us briefly discuss other qualitative indicators of the NOCE boundary, which could be found in the SABER database. As mentioned above, Kulikov et al. (2019) showed that the nighttime O SABER profiles are correct above the NOCE boundary, whereas the H profiles hold within the whole pressure interval. Kulikov et al. (2021) demonstrated that, in the altitude range of 80-85 km, many H profiles have a sharp jump in concentration when it increases from $\sim 10^7$ cm$^{-3}$ to $\sim 10^8$ cm$^{-3}$. Our analysis with the criterion (9) shows that the altitude of these jumps can be used as a rough indicator of the NOCE boundary."

There are quite a large number of figures (20) with different number of panels (from 1 to 20) in the article with a relatively small volume of text. For better structuring of the article, some of the figures should be omitted or merged, for example: (4, 8, 13 and 17), (6 and 15), (7 and 16), (10, 12, 19 and 20).

In the revised manuscript, the Figures were reorganized according to the Referee note.

Figures 1-3 show variations of O and H normalised to some mean daily values. It is not clear what these values are. If these are averages over the entire range of altitudes, then the figures should show known maximums of O and H, but this is not present in the figures. Apparently different daily average O and H values were used for each altitude. Please clarify this issue. Also, the figures show white spots where the normalised concentrations appear to fall below 10-6. Apparently the range of variation shown needs to be increased.

We have added the necessary clarification to the revised manuscript (see lines 157-159):

"In order to focus attention on diurnal oscillations, the concentrations are normalized by mean daily values, which were calculated as a function of altitude. These daily average O and H values were different for each altitude."

Also, the Figures were corrected according the Referee note.

Lines 315-317. The authors write "Basing on daytime O and H distributions in the mesopause region obtained in Kulikov et al. (2022), we calculated O/H in summer and winter." It would be nice to provide figures, which confirm that "this ratio at middle latitudes is remarkably less than in winter".

In the revised manuscript, we have added new Figure (Figure 14), which confirms this statement.

I have my doubts that quite a few of the many instances in which the articles are mentioned correctly. Furthermore, there are a number of questions about the use of English expressions. Therefore I strongly recommend checking the text of the article with the help of a professional translator.

The revised manuscript was verified and corrected by a professional translator.

Other changes are related to the recommendations and demands of other referee.

Thank you for taking your time to review our manuscript.

With respect,

Michael Kulikov, Michael Belikovich, Alexey Chubarov, Svetlana Dementyeva, and Alexander Feigin

---

## Author Comment (AC2)

Dear Editor,

We would like to say many thanks the Referee for taking the time to review our manuscript and providing valuable recommendations. Their constructive criticism made the work clearer and more precise. We took into account all the remarks of Referee and, to the best of our ability, implemented the corresponding changes in the manuscript.

In the following, we address the comments point by point and show how the manuscript has been changed according to the comments. Below we use a certain color notation: comments by Referee are in red, our responses are in black, and the changes in the manuscript are in blue (placed inside the quotation marks).

**Response to the comments on the paper by Referee 2**

This paper presents a derivation of the nighttime ozone chemical equilibrium (NOCE) condition. The motivation for the paper is that minor atomic species O and H are key elements of the energy budget and chemistry of the mesopause region. These species have no readily/easily observable features to enable their direct observation by satellites or from the ground. Consequently, the approach used to derive these species is to infer them based on chemical relations between species such as ozone and observations of radiative emissions from the hydroxyl radical. The assumption behind these approaches is that ozone and the species H and O are in chemical equilibrium with each other at night. The paper derives conditions under which the equilibrium assumptions would be true and assigns the upper altitude limit of these as the 'NOCE boundary.' The NOCE boundary is derived in depth based on observations made by the SABER instrument for the 20-plus year length of the mission. The NOCE boundary is analyzed in terms of altitude, latitude, and time. The NOCE boundaries are also examined for correlation with the 11-year solar cycle and for trends.

The paper presents interesting considerations for the derivation of O and H in the mesopause region from satellite observations. These considerations are important for the ongoing SABER instrument on the TIMED satellite and for analysis of data from the SCHIAMACHY instrument that formerly operated on EnviSAT.

Although the passage above contains not a comment but the summary of the paper we took liberty on commenting themselves. It was our oversight to omit other applications of the nighttime ozone chemical equilibrium (NOCE) condition except for O and H derivation. In the revised manuscript, we added in Discussion a few sentences in this sense:

"Note that the NOCE condition was used not only for O and H derivation from satellite data. This assumption is a useful approach helping (i) to study hydroxyl emission in the MLT region with simulated and measured data, in particular, OH* mechanisms, morphology and variability caused,

for example, by atmospheric tides and gravity wave activity (e.g., Marsh et al., 2006; Nikoukar et al., 2007; Xu et al., 2010, 2012; Kowalewski et al., 2014; Sonnemann et al., 2015); (ii) to analyze the MLT response to sudden stratospheric warmings (SSWs) (e.g., Smith et al., 2009); (iii) to derive exothermic heating rates of MLT (e.g., Mlynczak et al., 2013b); (iv) to analytically simulate the mesospheric OH* layer response to gravity waves (e.g., Swenson and Gardner, 1998); and (v) to derive the analytical dependence of excited hydroxyl layer number density and peak altitude on atomic oxygen and temperature (e.g., Grygalashvyly et al., 2014; Grygalashvyly, 2015)."

Also, we would like to emphasize other finding of the paper that the NOCE boundary is a marker of fundamental property of the $O_x$-$HO_x$ photochemistry in the MLT region: the NOCE boundary well reproduces the transition zone dividing deep and weak diurnal oscillations of O and H (see Figures 1-3). In the revised manuscript, we verified this feature with the annual run of SD-WACCM-X model for the year 2017. In Discussion, we added following paragraph and additional Figure:

"According to the used chemical-transport model, the NOCE boundary reproduces well the transition zone dividing deep and weak diurnal oscillations of O and H (see Figures 1-3). We verified this feature with the annual run of SD-WACCM-X model for the year 2017 provided by the NCAR High Altitude Observatory (https://doi.org/10.26024/5b58-nc53). Despite the low time resolution of the downloaded data (3-hour averaging), we obtained the results (see Figure 13) similar to Figures 1-3. Note also that both models give the same consistence between the altitudes of the NOCE boundary and the mentioned transition zone at high latitudes in spring and autumn."

Recommendation

The paper is returned for major revision. Below in the "Comments" section I list a number of comments for the authors to address. These are in no order of importance but rather they are chronological according to the paper.

There is one major weakness that I have for the entire paper – despite all the analyses of the NOCE boundary, and the apparent demonstration that ozone is not in chemical equilibrium over much of the mesopause region, the paper makes no statement as to how current the assumption of chemical equilibrium by SABER and SCHIAMCAHY affects the quality and the uncertainty in those datasets. Do they need reprocessing? How would that be done? What should current users assume for their analyses using these data? A revised paper must answer these questions.

Our main concern is that data users should be aware about NOCE issues. We believe that marking up the data according the proposed criteria would benefit the data users. We believe this comment is fully addressed in Discussion of the revised manuscript:

"The NOCE boundary is an important technical characteristic for correct application of the NOCE approximation to retrieve the nighttime distributions of minor chemical species of MLT. Kulikov et al. (2019) repeated the O and H retrieval by Mlynczak et al. (2018) from the SABER data for the year 2004. It was revealed that the application of the NOCE condition below the boundary obtained according to the criterion could lead to a great (up to 5–8 times) systematic underestimation of O concentration below 86 km, whereas it was insignificant for H retrieval. The results presented in Figures 4, 5 and 11 demonstrate that, except for high northern latitudes, there is a stable annual cycle of the NOCE boundary. The monthly mean boundary can rise up to geometrical altitudes of 82-83 km (~(5.2-6.2)·$10^{-3}$ hPa) at low latitudes and up to 84-85 km (~(3.7-4.4)·$10^{-3}$ hPa) at middle and high latitudes. Thus, the SABER O data below these altitudes/pressures may be essentially incorrect and the retrieval approaches without using the NOCE condition (e.g., Panka et al., 2018) should be more appropriate.

Note that the NOCE condition was used not only for O and H derivation from satellite data. This assumption is a useful approach helping (i) to study hydroxyl emission in the MLT region with simulated and measured data, in particular, OH* mechanisms, morphology and variability caused, for example, by atmospheric tides and gravity wave activity (e.g., Marsh et al., 2006; Nikoukar et al., 2007; Xu et al., 2010, 2012; Kowalewski et al., 2014; Sonnemann et al., 2015); (ii) to analyze the MLT response to sudden stratospheric warmings (SSWs) (e.g., Smith et al., 2009); (iii) to derive exothermic heating rates of MLT (e.g., Mlynczak et al., 2013b); (iv) to analytically simulate the mesospheric OH* layer response to gravity waves (e.g., Swenson and Gardner, 1998); and (v) to derive the analytical dependence of excited hydroxyl layer number density and peak altitude on atomic oxygen and temperature (e.g., Grygalashvyly et al., 2014; Grygalashvyly, 2015). Perhaps some results require revision or reanalysis taking the NOCE boundary into account. For example, Smith et al. (2009) used the NOCE condition to analyze the ozone perturbation in the MLT, in particular, during the SSW at the beginning of 2009 (the central day was January 24). Our preliminary results of processing the SABER and simulated data in January 2009 show that the NOCE boundary above 70ºN may jump from ~80 km to ~90-95 km due to a short-time abrupt temperature fall above 80 km during this SSW. Thus, one can assume that the NOCE condition is not a good approximation for the description of ozone variations directly in the process of SSWs. This case will be studied in a separate work."

Another major weakness in the paper is the trend analysis presented in Figures 12 and 20. These and their accompanying analyses and discussions must be deleted. The uncertainties (1-sigma) are larger than the derived trend, indicating that the trend is insignificant.

In the revised manuscript, these Figures were deleted and accompanying analyses were shortened.

Also, the distinction between zeq and zeqpa is not clear nor is the approach for deriving zeqpa. The paper needs to examine and clarify if these really are different parameters and why they both need to be analyzed to understand NOCE from a perspective of retrieving O and H from satellite observations.

In the revised manuscript, the corresponding section was rewritten including the justification for the importance of geometrical and pressure coordinates in analysis of long-term evolution:

"The total range of latitudes according to the satellite trajectory over a month was ~(83.5$^{\circ}$S - 83.5$^{\circ}$N). This range was divided into 20 bins and all local values of $p_{eq}{}^{l}$ and $z_{eq}{}^{l}$ falling into one bin during a month or a year were averaged, respectively. In particular, several thousand values of $p_{eq}{}^{l}$ and $z_{eq}{}^{l}$ fall into one bin during a month. Following Mlynczak et al. (2013a), averages were determined by binning the data of a certain day by local hour and then averaging over the hour bins that contain data to obtain the daily average value. Then we calculated monthly mean values of $p_{eq}{}^{m}$ and $z_{eq}{}^{m}$ and annually mean values of $p_{eq}{}^{y}$ and $z_{eq}{}^{y}$ (hereafter, the indexes «m» and «y» indicate the monthly and annually average, respectively). Then, for convenience, the values of $p_{eq}{}^{m}$ and $p_{eq}{}^{y}$ were recalculated into the pressure altitudes $h_{eq}{}^{m}$ and $h_{eq}{}^{y}$. The dependence of $h_{eq}{}^{m,y}$ on $p_{eq}{}^{m,y}$ was adopted from Mlynczak et al. (2013a, 2014):

$$h_{eq}{}^{m,y} = -H_a \cdot \log(p_{eq}{}^{m,y}/p_0), \ H_a = 5.753474, \ p_0 = 11430.49428 \text{ hPa.} \qquad (10)$$

Note that the use of both, geometrical and pressure coordinates is a rather common approach when analyzing long-term evolution of the obtained data, especially, when the data is the result of averaging over time and space. In particular, Lübken et al. (2013) demonstrated the importance of distinguishing between trends on pressure and geometrical altitudes in the mesosphere, since the second includes the atmospheric shrinking effect and is more pronounced. Grygalashvyly et al. (2014) analyzed the linear trends in OH* peak height and revealed a remarkable decrease at geometrical altitudes, which is almost absent at pressure altitudes."

Comments
Line 40 – I believe the authors intend the word 'reach' instead of 'rich'.
Corrected. See line 40 in the "Manuscript with changes incorporated".

Line 93-94 – It seems a word is missing between "according" and "the mentioned". Perhaps the authors mean "according to the mentioned"?
Corrected. See line 94 in the "Manuscript with changes incorporated".

Line 120-124. The authors neglect loss of ozone by reaction with atomic oxygen in the mesopause region. At the level of which they appear to be investigating the chemistry, this process should be included. Table 1 confirms that this reaction is not considered.

This reaction becomes important above ~ 95 km (Smith et al., 2009). Kulikov et al. (2023) verified with simulated (see Fig. 1 in Kulikov et al., 2023) and measured data that this reaction does not influence the NOCE boundary determination and can be skipped. In particular, it was found that that the ratio of the main ozone sink ($H+O_3 \rightarrow O_2+OH$) to the sink due to reaction $O + O_3 \rightarrow 2O_2$ is more than 50 near the NOCE boundary.

**References:**

Smith, A. K., Lopez-Puertas, M., Garcıa-Comas, M. and Tukiainen, S.: SABER observations of mesospheric ozone during NH late winter 2002–2009, Geophys. Res. Lett., 36, L23804, https://doi.org/10.1029/2009GL040942, 2009.

Kulikov, M. Yu., Belikovich, M. V., Chubarov, A. G., Dementeyva, S. O., Feigin, A. M.: Boundary of nighttime ozone chemical equilibrium in the mesopause region: improved criterion of determining the boundary from satellite data, Adv. Space Res., 71 (6), 2770-2780, https://doi.org/10.1016/j.asr.2022.11.005, 2023.

The manuscript was corrected. See lines 121-124 in the "Manuscript with changes incorporated":

« The secondary ozone loss via the $O + O_3 \rightarrow 2O_2$ reaction becomes important above ~ 95 km (Smith et al., 2009). Kulikov et al. (2023) verified with simulated and measured data that this reaction does not influence the NOCE boundary determination and may be skipped.»

Line 146-148. The authors need to specify which 'nighttime data' need to be 'excluded from consideration". O3, H, O, all?

Sorry for misunderstanding. This part was rewritten. See lines 149-153 in the "Manuscript with changes incorporated":

« The ozone equilibrium concentration jumps at sunset due to the shutdown of photodissociation. Thus, the condition (6) shows that it takes time for the ozone concentration to reach a new equilibrium. Kulikov et al. (2023) revealed that, at the solar zenith angle $\chi > 95°$, the condition (6) is fulfilled almost in all cases and the condition (5) becomes the main criterion for NOCE validity.»

Figures 1-3. The magenta line is very difficult to see in the figures unless they are substantially enlarged on the screen. It is also confusing with the large areas of similar colors above about 85 km in all figures. Perhaps a black line would be more visible? The caption should also state the meaning of the $Cr = 0.1$ condition/boundary to facilitate the interpretation of the figure.

We have changed the figures following the Referee suggestions. The captions were corrected accordingly. Magenta lines in Figures 1-3 were replaced by black lines, see, for example, Figure 1 from the "Manuscript with changes incorporated":

[Figure]

Lines 170-185. The analysis here must be re-worked and compared with the model and derivation of O and H reported by Panka et al., including the results of H and OH reported in the literature. To date there has been no extensive comparison of the Panka et al. [O] results with the Mlynczak et al.,

2018 results. The Panka et al. approach appears to more rigorously include the recent discovery of the importance of collisions with O(1D). The Panka data are hosted on the SABER website. https://agupubs.onlinelibrary.wiley.com/doi/10.1029/2018GL077677 https://agupubs.onlinelibrary.wiley.com/doi/10.1029/2020GL091053

We find this comment open to interpretation. In the pointed section, the results from Kulikov et al. (2018a) are used to derive the NOCE criterion (5) in the form of the expression containing only measurable characteristics. (Note Kulikov et al. (2023) demonstrated using chemical-transport model that the criterion (5) almost ideally reproduces the NOCE boundary, see Figure 1 in Kulikov et al. (2023)). For this, the equations of the physicochemical balance of excited OH at levels 8 and 9 are used, where the reaction H + $O_3$ is the source of these exited states. The lifetimes of OH(9) and OH(8) are much less than 1 sec, so there is no doubt that the conditions for their equilibrium are fulfilled. Here, we use the OH(v) model by Mlynczak et al. (2013, 2018) which includes all main processes to determine the OH(9) and OH(8) equilibrium concentration: formation via the reaction $H+O_3 \rightarrow OH(v)+O_2$, spontaneous emissions due to transitions 9→8, 9→7, 8→6, and the quenching by O, $O_2$ and $N_2$. However, the OH(9) and OH(8) equilibrium concentration and final expression for the NOCE criterion are determined by a number of parameters which, of course, may not be accurate.

As we understand, the reviewer suggests considering Panka et al. data instead of original SABER data. We downloaded and analyzed dataset by Panka et al. from the SABER website. It was revealed that there are following difficulties to use these data in the goals of the paper:

(1) The papers by Panka et al. (2018, 2020) did not retrieve the atomic hydrogen. So, there is no H in the Panka data on the SABER website. Note H data are needed for our criterion (5).

(2) There is no $O_3$ in the Panka dataset on the SABER website. Note $O_3$ data are also needed for criterion (5).

(3) The latter point can be remedied by collocation of Panka data with original SABER data, but it requires some additional guesswork as not all processing details are revealed in the corresponding papers. Which in turn devalues the credibility of the results to be obtained. In particular, the vertical resolution of the Panka data is ~1 km that is in ~3 times worse than the resolution of SABER data. The averaging and interpolation procedures used in the papers are not described explicitly. Additionally, 1 km resolution is not good in sense of uncertainty in determining the NOCE boundary, which is mainly located in a narrow altitude range of 80-85 km according to our results. Moreover, the altitude range of the Panka data is limited by 80-100 km. Thus, when the NOCE boundary altitude is less 80 km, its determination is impossible.

(4) The O retrieval procedure proposed in Panka et al. (2018) is based on the approach that the ratio between the volume emission rates measured at 2.05 and 1.6 μm ($VER_2/VER_{1.6}$) is proportional to

O volume mixing ratio (O VMR). Panka et al. (2018) demonstrated it on the Figure 1 at 85, 90, 95, 100 km (see right panel below):

[Figure]

**Figure 1.** (left) Volume emission rate dependence on atomic oxygen density for SABER channel 8 and channel 9 at four different pressure levels. (right) Ratios of volume emissions rates (Ch8/Ch9) at the respective pressure levels. Red - 2.4e-7 atm (scaled by 0.125); Green - 5.7e-7 atm (scaled by 0.25); Blue - 1.5e-6 atm (scaled by 0.5); Green - 4.0e-6 atm (not scaled). Solid - OH non-LTE model with reaction (2); Solid with circles - OH non-LTE model without reaction (2).

This Figure was copied from Panka et al. (2018). Note that the x-axis in the right panel does not start at 0, but at a value of $\sim 10^{-3}$.

However, one can see from the Figure that the dependence of $VER_2/VER_{1.6}$ on the O VMR at 85 km (see yellow line without circles and straight dotted line near it, the latter is included by us) becomes to be nonlinear at low values of O VMR below $\sim(2\text{-}3)\cdot10^{-3}$. It is well-known, that O VMR below 90 km decreases rapidly with height loss due to decreasing O absolute value and increasing air concentration, so O VMR can be several orders of magnitude lower than $(2\text{-}3)\cdot10^{-3}$. It can be seen in following Figure, which presents the range of O VMR variation at different altitudes. The data were taken directly from the Panka dataset. Here, as an example, we show all O VMR values (blue dots) corresponding to 2 January 2009.

[Figure]

One can see, at 85 km, main part of the values are less than $(2\text{-}3)\cdot 10^{-3}$. So, one may question the validity of Panka et al. approach at altitudes of 80-85 km.

Thus, all things considered, we can't see the Panka data particularly suitable for determination of NOCE boundary altitude.

**References:**

Kulikov, M. Y., Belikovich, M. V., Grygalashvyly, M., Sonnemann, G. R., Ermakova, T. S., Nechaev, A. A., and Feigin, A. M.: Nighttime ozone chemical equilibrium in the mesopause region. J. Geophys. Res.,123, 3228–3242, https://doi.org/10.1002/2017JD026717, 2018.

Kulikov, M. Yu., Belikovich, M. V., Chubarov, A. G., Dementeyva, S. O., Feigin, A. M.: Boundary of nighttime ozone chemical equilibrium in the mesopause region: improved criterion of determining the boundary from satellite data, Adv. Space Res., 71 (6), 2770-2780, https://doi.org/10.1016/j.asr.2022.11.005, 2023.

Mlynczak, M. G., Hunt, L. A., Mast, J. C., Marshall, B. T., Russell III, J. M., Smith, A. K., Siskind, D. E., Yee, J.-H., Mertens, C. J., Martin-Torres, F. J., Thompson, R. E., Drob, D. P., and Gordley, L. L.: Atomic oxygen in the mesosphere and lower thermosphere derived from SABER: Algorithm theoretical basis and measurement uncertainty, J. Geophys. Res., 118, 5724–5735, https://doi.org/10.1002/jgrd.50401, 2013a.

Mlynczak, M. G., Hunt, L. A., Russell, J. M. III, and Marshall, B. T.: Updated SABER night atomic oxygen and implications for SABER ozone and atomic hydrogen, Geophys. Res. Lett., 45, 5735–5741, https://doi.org/10.1029/2018GL077377, 2018.

Panka, P. A., Kutepov, A. A., Rezac, L., Kalogerakis, K. S., Feofilov, A. G., Marsh, D., et al. (2018). Atomic oxygenbretrieved from the SABER 2.0- and 1.6-μm radiances using new first-principles nighttime OH(v) model. Geophysical Research Letters, 45, 5798–5803. https://doi.org/10.1029/2018GL077677

Panka, P. A., Kutepov, A. A., Zhu, Y., Kaufmann, M., Kalogerakis, K. S., Rezac, L., et al. (2021). Simultaneous retrievals of nighttime O(3P) and total OH densities from satellite observations of Meinel band emissions. Geophysical Research Letters, 48, e2020GL091053. https://doi.org/10.1029/2020GL091053

Line 191 – First full sentence, "In the second case.."

Corrected. See line 198 in the "Manuscript with changes incorporated".

Line 194-198, and the subsequent description of Figures 4, 5, 6, and 7. The exact altitude of each pressure level is included in the SABER database. There is no need to approximate these altitudes based on a small subset of approximate altitudes and pressures mentioned in the Mlynczak 2013a and 2014 references. These altitudes are derived by SABER consistent with hydrostatic balance and the very accurate pointing of the SABER instrument. As presented in the paper, there could be substantial error in the assignment of altitude based on the method of altitude assignment as described by the authors.

The confusion is caused by the description, which, upon reflection, was not entirely correct. We determine the local position of the NOCE boundary as pressure level $p_{eq}$ and altitude level $z_{eq}$, respectively. Then, all local values of $p_{eq}$ and $z_{eq}$ falling into one latitude bin during a month are averaged. Only after averaging, monthly mean $p_{eq}$ is recalculated into the pressure altitude, for convenience. This section has been rewritten including justification for the importance of geometrical and pressure coordinates in the analysis of long-term evolution in obtained data (see lines 200-217):

"The total range of latitudes according to the satellite trajectory over a month was ~(83.5$^{\circ}$S - 83.5$^{\circ}$N). This range was divided into 20 bins and all local values of $p_{eq}{}^{l}$ and $z_{eq}{}^{l}$ falling into one bin during a month or a year were averaged, respectively. In particular, several thousand values of $p_{eq}{}^{l}$ and $z_{eq}{}^{l}$ fall into one bin during a month. Following Mlynczak et al. (2013a), averages were determined by binning the data of a certain day by local hour and then averaging over the hour bins that contain data to obtain the daily average value. Then we calculated monthly mean values of

$p_{eq}{}^m$ and $z_{eq}{}^m$ and annually mean values of $p_{eq}{}^y$ and $z_{eq}{}^y$ (hereafter, the indexes «m» and «y» indicate the monthly and annually average, respectively). Then, for convenience, the values of $p_{eq}{}^m$ and $p_{eq}{}^y$ were recalculated into the pressure altitudes $h_{eq}{}^m$ and $h_{eq}{}^y$. The dependence of $h_{eq}{}^{m,y}$ on $p_{eq}{}^{m,y}$ was adopted from Mlynczak et al. (2013a, 2014):

$$h_{eq}{}^{m,y} = -H_a \cdot \log\left(p_{eq}{}^{m,y}/p_0\right), \quad H_a = 5.753474, \quad p_0 = 11430.49428 \text{ hPa.} \qquad (10)$$

Note that the use of both, geometrical and pressure coordinates is a rather common approach when analyzing long-term evolution of the obtained data, especially, when the data is the result of averaging over time and space. In particular, Lübken et al. (2013) demonstrated the importance of distinguishing between trends on pressure and geometrical altitudes in the mesosphere, since the second includes the atmospheric shrinking effect and is more pronounced. Grygalashvyly et al. (2014) analyzed the linear trends in OH* peak height and revealed a remarkable decrease at geometrical altitudes, which is almost absent at pressure altitudes."

Additionally, below are the Figures from the papers by Lübken at al. (2013) and Grygalashvyly at al. (2014), that demonstrate «It is important to distinguish between trends on pressure altitudes, $z_p$, and geometrical altitudes, $z_{geo}$, where the latter includes the effect of shrinking due to cooling at lower heights.» as noted by Lübken at al. (2013).

[Figure]

**Figure 5.** Temperature variations in the latitude band 45°N–55°N on a (left) pressure and (right) geometrical altitude of 70 km for run 7. LIMA temperatures (black lines) are compared with a multivariate fit (red lines) consisting of carbon dioxide, ozone, and $Ly_\alpha$ variations as shown in Figure 3. For comparison, a linear fit over the entire time period is also shown (green line). The orange curve shows the variation of $Ly_\alpha$ and the cyan curve the variation of ozone (right axis; see text for more details). The two squares mark temperatures in 1975 and 1976 which are ignored when fitting (see text).

This Figure was copied from Lübken at al. (2013).

[Figure]

**Figure 9.** (a, b) Long-term variations (dashed lines) and corresponding linear trends (solid lines) of $OH^*_{v=6}$ height in pressure and geometrical coordinates for winter (blue line), spring (green line), and summer (red line) at 51.25°N.

This Figure was copied from Grygalashvyly at al. (2014).

Line 198. The authors should also specify, again to remind the reader, the meaning of peq and zeqpa, and state this in the Figure captions.

Proper reminder is present in revised manuscript.

Figure 6. The comparisons shown are for calendar months. The paper mentions 'changes in satellite geometry'. These 'changes' are such that the local time sampled by the spacecraft (and 3 hence, by SABER) is does not remain constant in a given calendar month over the course of the mission. It is slowly drifting. Thus, the local times sampled in every "January" are not the same over the course of the mission. Please see:

https://agupubs.onlinelibrary.wiley.com/doi/10.1029/2018JA025892

This fact suggests that the results in Figure 6 and any other Figure or results involving monthly calculations or comparisons may be incorrect.

In the discussion version of the manuscript, we averaged the data as a whole, i.e., without time binning. In the revised manuscript, we recalculated all data with binning on days and local hours, following to Mlynczak et al. (2013a). The new Figures slightly changed from previous, as it can be seen below:

[Figure]

Red curve: $F_{10.7}$ index (solar radio flux at 10.7 cm). Blue curve: latitude-averaged pressure altitude $h_{eq}{}^{y}$ in the range between ~55ºS and ~55ºN. Black curve is "old" variant of the blue one.

The change in the result is negligible. In the revised manuscript, we added an indication of how the averaging was performed (see lines 203-207):

"Following Mlynczak et al. (2013a), averages were determined by binning the data of a certain day by local hour and then averaging over the hour bins that contain data to obtain the daily average value. Then we calculated monthly mean values of $p_{eq}{}^{m}$ and $z_{eq}{}^{m}$ and annually mean values of $p_{eq}{}^{y}$ and $z_{eq}{}^{y}$ (hereafter, the indexes «m» and «y» indicate the monthly and annually average, respectively)."

Figure 7. Please mark the periods less than one year on the x-axis to make it easier for the reader to discern these. What is the statistical significance of these features? Have the authors considered a Lomb periodogram which provides a significance test for the derived periodic features?
mark the periods less than one year on the x-axis

In the revised manuscript, we added the periods less than one year on the x-axis. There was a misunderstanding here, perhaps, due to an insufficiently correct description. These spectra are usual spectra obtained from the time evolution of monthly mean pressure altitude at certain latitudes shown on previous Figure 6. So, additional clarification of term "statistical significance" is needed in this context. The Lomb–Scargle periodogram is a well-known method for detecting and characterizing periodic signals in unevenly sampled data. We did not consider Lomb periodogram, because, in our case, analyzed time series are uniform (12 points for the each year).

To avoid possible misunderstanding, in the revised manuscript, corresponding sentences were rewritten (see lines 226-229):

" Figure 4 demonstrates the time evolution of the pressure altitude $h_{eq}{}^{m}$ in 2002-2021 in all latitude bins. Figures 5 (left column) show the mean (for 2002-2021) annual cycle of $h_{eq}{}^{m}$ at four specific latitudes and Figures 6 (left column) present the Fourier spectra at these latitudes obtained from the data in Figure 4."

Lines 234-243. The paper describes anticorrelation between average zeqpa and the 11-year solar cycle. The authors should provide some rationale for why the altitude/pressure at which night ozone is (or is not) in photochemical equilibrium depends on the slowly varying solar cycle.
Specifically, what atmospheric characteristics cause this? Is it a feedback from temperature and perhaps the 'breathing' or expansion and contraction of the atmosphere with the cyclic heating and cooling of the solar cycle?

In order to answer these questions, we added in Discussion of the revised manuscript the following text (see lines 342-361):

"Thus, at low and middle latitudes $h_{eq}{}^{y}$ can be considered as a sensitive indicator of solar activity. Below, we present a simple and short explanation for this. Let us consider the NOCE criterion (9) at the pressure level $p_{eq}$:

$$VER(p_{eq}) = VER_{min}\left(T, M(p_{eq})\right).$$

In a zero approximation

$$VER_{min} = 20 \cdot \frac{k_1 \cdot O_2(p_{eq}) \cdot M(p_{eq}) \cdot \left(k_4 \cdot O_2(p_{eq}) \cdot M(p_{eq}) \cdot \left(1 - \frac{k_5+k_6}{k_3}\right) + k_2 \cdot O_3(p_{eq})\right) \cdot A(T, M(p_{eq}))}{k_2} \cong 20 \cdot$$

$$\frac{k_1 \cdot k_4 \cdot (O_2(p_{eq}) \cdot M(p_{eq}))^2 \cdot A(T, M(p_{eq}))}{k_2} \sim \frac{k_1 \cdot k_4 \cdot (p_{eq}/T)^4 \cdot A(T, p_{eq})}{k_2} \sim \frac{exp(470/T) \cdot p_{eq}^4 \cdot A(T, p_{eq})}{T^{8.2}},$$

where $A(T, p_{eq}) =$

$$\frac{0.47 \cdot 118.35}{215.05 + 2.5 \cdot 10^{-11} \cdot O_2 / M \cdot \frac{p_{eq}}{k_B T} + 3.36 \cdot 10^{-13} \cdot e^{\frac{220}{T}} \cdot N_2 / M \cdot \frac{p_{eq}}{k_B T}} + \frac{0.34 \cdot 117.21}{178.06 + 4.8 \cdot 10^{-13} \cdot O_2 / M \cdot \frac{p_{eq}}{k_B T} + 7 \cdot 10^{-13} \cdot N_2 / M \cdot \frac{p_{eq}}{k_B T}} +$$

$$\frac{0.47 \cdot 117.21 \cdot (20.05 + 4.2 \cdot 10^{-12} \cdot O_2 / M \cdot \frac{p_{eq}}{k_B T} + 4 \cdot 10^{-13} \cdot N_2 / M \cdot \frac{p_{eq}}{k_B T})}{(215.05 + 2.5 \cdot 10^{-11} \cdot O_2 / M \cdot \frac{p_{eq}}{k_B T} + 3.36 \cdot 10^{-13} \cdot e^{\frac{220}{T}} \cdot N_2 / M \cdot \frac{p_{eq}}{k_B T}) \cdot (178.06 + 4.8 \cdot 10^{-13} \cdot O_2 / M \cdot \frac{p_{eq}}{k_B T} + 7 \cdot 10^{-13} \cdot N_2 / M \cdot \frac{p_{eq}}{k_B T})}.$$

Our analysis of $A(T, p_{eq})$ shows that this function can be approximately rewritten as $A(T, p_{eq}) \approx const + \frac{const}{const + \frac{p_{eq}}{T}}$. So, one can see that $VER_{min}$ is strongly dependent on $T$. Moreover, it anticorrelates with $T$. Gan et al. (2017) and Zhao et al. (2020) analyzed the simulated and measured data and revealed a clear correlation between the MLT temperature above 80 km and the 10.7-cm solar radio flux. Moreover, the dependence of the correlation coefficient of $T$ with $F_{10.7}$ index on latitude in the 55ºS-55ºN range given in Figure 9 in the paper by of Zhao et al. (2020) is consistent with our Figure 9 (left panel), taking into account the sign of the correlation. Thus, we can conclude that the found anticorrelation of the NOCE boundary $h_{eq}{}^y$ with solar activity is caused by the strong connection with temperature, which, in turn, is in a good correlation with the $F_{10.7}$ index. A detailed analysis of the reasons why the solar cycle weakly manifests itself in the spatio-temporal variability of $z_{eq}{}^y$ is not so simple and is beyond the scope of this work. ''

Line 244-249. It is assumed that the uncertainties plotted in Figure 12 are 1-sigma values. All of the 1-sigma uncertainties are larger than the derived trend value, except at 20 N. At 2-sigma, all of them are. These results and figure 11 should be removed from the paper as the significance of them is marginal at best.

In the revised manuscript, the Figures were deleted and accompanying analysis was shortened.

Figure 13. The color bar is very difficult to read as the altitudes all run together. It is also very difficult to discern anything quantitative about the annual variations shown in the figure.

In the revised manuscript, this Figure was deleted.

Figure 14. Although there are differences in Figures 5 and 14 (zeqpa and zeq, respectively), it is not clear what these differences are trying to show, or if there is a real difference. See my previous

comment about the assignment of altitude to zeqpa. For clarity, SABER's natural vertical coordinate is pressure. All the data are retrieved as a function of pressure. There is no 'assignment' of pressure. Similarly, the temperature retrieval profile that is in hydrostatic equilibrium, the altitudes assigned to the pressure surfaces are accurate, and are derived in part from the accurate knowledge of the position of the field of view of the instrument as is scans the limb.

The SABER database gives altitudes and pressures, as noted in Mlynczak et al. (2013):

**1.1. Data Preparation and Screening**

[8] All of the algorithms and results reported here are for version 1.07 of the SABER data set and will be used in the upcoming version 2 of the data set. Profiles of pressure, altitude, temperature, ozone volume mixing ratio (derived from the 9.6 μm channel), and OH volume emission rate (at 2.0 μm) are extracted from the SABER level 2A data file. A small number of profiles are rejected if more than

The investigation of the means of how the how altitude and pressure values were obtained originally lays beyond the scope of current paper. We use both characteristics provided by SABER database. We observe the difference depending the use of pressure altitude or geometrical altitude. The reason, why the difference takes place, is described above. To avoid possible misunderstanding, in the revised manuscript, corresponding sentence was rewritten (see lines 174-177):

"We use version 2.0 of the SABER data product (Level2A) for the simultaneously measured profiles of pressure ($p$), altitude ($z$), temperature (T), $O_3$ (at 9.6 μm), and total volume emission rates of OH* transitions at 2.0 ($VER$) within the 0.0001–0.02 mbar pressure interval (altitudes approximately 75–105 km) in 2002-2021."

Figures 17-18. Same comment as for Figure 14.

See the reply on previous comment.

Figure 19 and 20. These results and Figures should be deleted. The trends in Figure 20 are not significant even at the 1-sigma level.

In the revised manuscript, the Figure 20 was deleted and accompanying analysis was shortened. We believe Figure 19 is still useful.

Line 290-297. The reason for the discrepancy noted in the zeq and zeqpa may have a simple explanation, as noted above. It is not clear how the authors assigned zeqpa, as noted above.

We believe the issue is already addressed in previous replies.

Other changes are related to the recommendations of other referee.

Thank you for taking your time to review our manuscript.

With respect,

Michael Kulikov, Michael Belikovich, Alexey Chubarov, Svetlana Dementyeva, and Alexander Feigin